# Antimicrobial Activity of Extracts from the *Humiria balsamifera* (Aubl)

**DOI:** 10.3390/plants10071479

**Published:** 2021-07-19

**Authors:** Edelson de J. S. Dias, Antônio J. Cantanhede Filho, Fernando J. C. Carneiro, Cláudia Q. da Rocha, Luís Cláudio N. da Silva, Joice C. B. Santos, Thayná F. Barros, Deivid M. Santos

**Affiliations:** 1Chemistry Graduate Program, Federal Institute of Education, Science and Technology of Maranhão, Avenida Getúlio Vargas 04, Monte Castelo, São Luís 65030-005, MA, Brazil; prof.antoniofilho@ifma.edu.br (A.J.C.F.); fernandocarneiro@ifma.edu.br (F.J.C.C.); 2Chemistry Graduate Program, Federal University of Maranhão, Av. dos Portugueses, 1966—Vila Bacanga, São Luís 65080-805, MA, Brazil; rocha.claudia@ufma.br; 3Graduate Program in Microbial Biology, CEUMA University, Rua Josué Montello, 1—Renascença II, São Luís 65075-120, MA, Brazil; luiscn.silva@ceuma.br (L.C.N.d.S.); joic.cast@hotmail.com (J.C.B.S.); thata.fernandesbarros@hotmail.com (T.F.B.); deivid.martinss98@gmail.com (D.M.S.)

**Keywords:** Humiriaceae, *Humiria*, antimicrobial, microorganisms, flavonoids

## Abstract

*Humiria balsamifera* (Aubl), commonly known as “mirim”, is a plant of the Humiriaceae family, which consists of 39 species divided between eight genera: *Duckesia, Endopleura, Humiria, Humiriastrum, Hylocara, Sacoglottis, Schistostemon,* and *Vantenea*. This study aimed to characterize *H. balsamifera* extracts by LC-MS/MS and evaluate their antimicrobial potential through in vitro and in vivo assays. The leaves and stem bark of *H. balsamifera* were collected and dried at room temperature and then ground in a knife mill. The extracts were prepared with organic solvents in order to increase the polarity index (hexane, ethyl acetate, and methanol). The antimicrobial effects of these extracts were evaluated against the following bacterial strains: *Escherichia coli ATCC 25922, Listeria monocytogenes* ATCC 15313, *Salmonella enterica* Typhimurium ATCC 14028, and *Staphylococcus aureus ATCC 6538*. The best activity was observed in the ethyl acetate (EALE = 780 µg/mL), methanol (MLE = 780 µg/mL), and hexane (HLE = 1560 µg/mL) leaf extracts against *S. aureus*. Considering the results for both antimicrobial and antibiofilm activities, the EALE extract was chosen to proceed to the infection assays, which used *Tenebrio molitor* larvae. The EALE treatment was able to extend the average lifespan of the larvae (6.5 days) in comparison to *S. aureus*-infected larvae (1 day). Next, the samples were characterized by High-Performance Liquid Chromatography coupled to a mass spectrometer, allowing the identification of 11 substances, including seven flavonoids, substances whose antimicrobial activity is already well-reported in the literature. The number of bioactive compounds found in the chemical composition of *H. balsamifera* emphasizes its significance in both traditional medicine and scientific research that studies new treatments based on substances from the Brazilian flora.

## 1. Introduction

Microorganisms are naturally well-spread out in the environment, and they can easily reach surfaces people come into contact with, including food products, whether at the harvest, slaughter, processing, or even packaging. Once in contact with the food, they start their growth process by consuming nutrients and causing the product to deteriorate [1,2,3].

Bacteria, fungi, viruses, and protozoa are the main microorganisms responsible for food contamination, infecting humans through the consumption of beef, fish, poultry, eggs, unhygienic fruits and fresh produce, causing a variety of diseases [4,5]. The World Health Organization estimates that one in 10 people worldwide become ill after consuming contaminated food and about 420,000 people die each year, resulting in the loss of 33 million healthy life years (DALYs) [2,6].

Bacteria represent an added concern for health and food safety organizations, especially those able to grow at low temperatures and resist a wide range of temperature variations [7,8]. Bacterial pathogens such as *Escherichia coli*, *Salmonella enterica*, *Listeria monocytogenes*, and *Staphylococcus aureus*, among others, are responsible for several global foodborne outbreaks and cause life-threatening illnesses such as diarrheal diseases [5,9,10,11,12]. Another problem in fighting bacteria is their ability to develop resistance to conventional antimicrobials. These pathogens can use various strategies to inhibit the effects of antimicrobials, such as the production of inactivating enzymes, reduction of outer membrane permeability, efflux system, and blocking or altering the target site of antibiotics, further motivating the research focused on finding alternative ways to combat them [13,14].

In the search for new effective substances against resistant pathogens, several secondary metabolites from plants and endophytic microorganisms have shown promise [15,16]. Most of the drugs used in general today were developed based on ethnopharmacological knowledge [17,18,19], indicating that the chemistry of natural products is a big ally in the development of therapeutic agents [20,21].

The plant species *Humiria balsamifera* (Aubl), popularly known as “mirim”, presents interesting biological activities. The literature reports, most of all, anti-inflammatory [22,23], antimalarial [24], antioxidant [25,26], and antifungal activity [27], highlighting the therapeutic potential of this plant. Some substances isolated from this species so far have already been reported as well, such as bergenin, arjunolic acid, friedelin, lupeol, phytol, caryophyllene oxide, epoxide humulene, and trans-isolongifolanone, among others [24]. However, the antibacterial and antibiofilm activities of its derived products have not been extensively examined. Thus, this work aims to characterize and evaluate the effectiveness of *H. balsamifera* extracts in terms of the antimicrobial and antibiofilm activities against foodborne pathogens (*Escherichia coli ATCC 25922, Listeria monocytogenes* ATCC 15313, *Salmonella*
*enterica* Typhimurium ATCC 14028, and *Staphylococcus aureus ATCC 6538*). The in vivo antimicrobial action of the most active extract was analyzed using a method based on the infection of *Tenebrio molitor* larvae.

## 2. Results

### 2.1. Antimicrobial Activity Evaluation

The antimicrobial activity of *Humiria balsamifera* (Aubl) leaf and stem bark extracts was evaluated by the determination of their minimum inhibitory concentrations (MIC) against four foodborne bacteria species: *E. coli*, *L. monocytogenes*, *S. enterica* Typhimurium, and *S. aureus* (Table 1).

Stem bark extracts did not exhibit antimicrobial action at any of the concentrations tested (MIC > 1250 µg/mL). However, the leaf extracts successfully inhibited *S. aureus*, with MIC = 780 µg/mL (EALE and MLE) and 1560 µg/mL (HLE). The EALE and the MLE also inhibited *L. monocytogenes* (MIC = 3120 µg/mL). The leaf extracts presented no action against the Gram-negative bacteria tested in this study.

### 2.2. Evaluation of the Antibiofilm Activity of Humiria balsamifera (Aubl) Extracts

Since the leaf extracts of *Humiria balsamifera* presented better inhibition results against *S. aureus*, their antibiofilm action at subinhibitory concentrations (0.5 × MIC, 0.25 × MIC, 0.125 × MIC, and 0.0625 × MIC) was also evaluated (Figure 1). The EALE and the HLE reduced biofilm production by *S. aureus* by nearly 25% at concentrations higher than 390 µg/mL (Figure 1A,B). The MLE did not exhibit significant antibiofilm activity (Figure 1C). Considering the results for the antimicrobial and antibiofilm activities, the EALE was chosen to proceed to the in vivo tests, using *Tenebrio molitor* larvae.

### 2.3. Evaluation of the In Vivo Activity of the Ethyl Acetate Leaf Extract of Humiria balsamifera (Aubl)

To evaluate the antimicrobial efficacy of the ethyl acetate leaf extract (EALE), the selected method was an alternative infection model based on the *S. aureus* ability to infect *T. molitor* larvae (Figure 2). The group infected with a lethal dose of *S. aureus* with no treatment presented an average lifespan of 1 day. In contrast, the uninfected larvae inoculated with the extract or its vehicle did not show a decrease in their lifespan. The EALE treatment was able to extend the average lifespan of the larvae (6.5 days), and by the end of the evaluation period, 50% of all larvae in this group were still alive.

### 2.4. Chemical Characterization of Humiria balsamifera (Aubl) Leaf and Stem Bark Extracts

Analyses of the leaf and stem bark extracts of *Humiria balsamifera* (Aubl) by HPLC-ESI-IT/MS in negative-ion mode identified 11 molecular ions (Table 2, Table 3 and Table 4). Their structures were proposed (Figure 3) based on the fragments originated from the molecular ion by multi-stage mass spectrometry (MS^n^). The mass spectrometry ionization source was the electrospray (ESI). The ESI source may not have ionized the compounds like steroids and triterpenes. It was possible to identify only phenolic compounds in the extracts. From the 11 identified substances, seven were flavonoids (gallocatechin, kaempferol 3-neohesperidoside, sophoricoside, quercetin 3-arabinoside, quercetin-*O*-rhamnoside, kaempferol-dirhamnoside, and vitexin-dirhamnoside); three were coumarins (bergenin and two derivatives: galloylbergenin and hydroxybenzoyl bergenin); and one was an oligosaccharide (maltotetraose).

Maltotetraose (1) presented a molecular ion of *m*/*z* = 665, with a fragment of *m*/*z* = 664 after the loss of a proton. In the third stage of the fragmentation process, the loss of 341 Da as C_12_H_21_O_11_ and 18 Da as a water molecule produced a fragment of *m*/*z* = 305.

Bergenin (2) originated four ionic fragments: from an initial loss of 60 Da as C_2_H_4_O_2_, and 18 Da as a water molecule, resulted the fragment C_12_H_10_O_6_^−^ (*m*/*z* = 249); the second one was produced from the loss of 93 Da as C_2_H_4_O_2_, a methyl group, and a water molecule, resulting in the ion C_11_H_7_O_6_^−^ (*m*/*z* = 234); the third fragment C_10_H_8_O_5_^−^ (*m*/*z* = 207) was the result of the loss of 120 Da as C_4_H_8_O_4_; and finally, the loss of 135 Da as C_4_H_8_O_4_ and a methyl group led to the fourth fragment, C_9_H_5_O_5_^−^ (*m*/*z* = 192).

All these ions were also present in the spectra of the bergenin derivatives, plus one other fragment, observed in both spectra.

The fragment C_14_H_15_O_9_^−^ (*m*/*z* = 327) was observed in both spectra—galloylbergenin (3) and hydroxybenzoyl bergenin (4). For galloylbergenin, this fragment was a result of the loss of 152 Da as the galloyl group. For hydroxybenzoyl bergenin, this fragment resulted from the elimination of the hydroxybenzoyl group (136 Da).

Gallocatechin (5) produced two fragment ions: C_9_H_8_O_4_^−^ (*m*/*z* = 179), resulting from the loss of 126 Da as C_6_H_6_O_3_, and C_8_H_8_O_4_ (*m*/*z* = 165) from the loss of 140 Da as C_7_H_8_O_3_.

Kaempferol 3-*O*-neohesperidoside (6) also produced only two fragment ions: C_21_H_19_O_10_^−^ (*m*/*z* = 431), generated by the loss of 162 Da as C_6_H_10_O_5_, and C_21_H_15_O_9_^−^ (*m*/*z* = 411), originated by the ion C_21_H_19_O_10_^−^, after a water loss. Bergenin (2) was also identified.

Sophoricoside (7) originated three ions: C_21_H_19_O_10_^−^ (*m*/*z* = 431), as a result of the loss of 90 Da as C_3_H_6_O_3_, C_17_H_11_O_6_^−^ (*m*/*z* = 311) from the loss of 120 Da as C_4_H_8_O_4_^−^, and from C_17_H_11_O_6_^−^, the third fragment, C_16_H_11_O_5_^−^ (*m*/*z* = 283), was formed as a result of the loss of 28 Da as a carbon monoxide molecule.

In the quercetin 3-arabinoside spectrum (8), four fragment ions were observed: C_15_H_9_O_7_^−^ (*m*/*z* = 300) from the loss of 133 Da as C_5_H_9_O_4_, C_14_H_7_O_6_^−^ (*m*/*z* = 271) from the loss of 162 Da as C_6_H_10_O_5_, and the fragments C_13_H_7_O_5_^−^ (*m*/*z* = 243) and C_13_H_7_O_4_^−^ (*m*/*z* = 227), resulting from the loss of a carbon monoxide and dioxide, respectively.

The fragmentation of quercetin-*O*-rhamnoside (9) were similar to what was observed for quercetin 3-arabinoside. The loss of a rhamnose molecule, C_6_H_11_O_4_ (*m*/*z* = 147 Da), led to the fragment ion C_15_H_9_O_7_^−^ (*m*/*z* = 300). Then, the loss of carbon monoxide produced the fragments C_14_H_7_O_6_^−^ (*m*/*z* = 271) and C_13_H_7_O_5_^−^ (*m*/*z* = 243).

The fragmentation of kaempferol-dirhamnoside produced four ions. The first one, C_21_H_19_O_10_^−^ (*m*/*z* = 431), was generated by the loss of a rhamnose molecule (C_6_H_11_O_4_, *m*/*z* = 147 Da). From the C_21_H_19_O_10_^−^ fragment (*m*/*z* = 431), the loss of a water molecule produced the second fragment ion, C_21_H_17_O_9_^−^ (*m*/*z* = 413), which, in turn, generated the fragment C_18_H_13_O_7_^−^ (*m*/*z* = 341) by the loss of 72 Da as C_3_H_4_O_2_. Finally, the loss of 41 Da as C_2_OH produced the fragment C_16_H_12_O_6_^−^ (*m*/*z* = 300).

The fragmentation of vitexin-dirhamnoside resulted in two ionic products: C_21_H_17_O_9_^−^ (*m*/*z* = 413), generated by the loss of a rhamnose molecule (C_6_H_11_O_4_, 147 Da), and C_17_H_9_O_5_^−^ (*m*/*z* = 293), produced by the loss of C_4_H_6_O_3_ (*m*/*z* = 102), followed by a water molecule.

## 3. Discussion

This research aimed to characterize and evaluate the antimicrobial potential of the extracts of *Humiria balsamifera* (Aubl), also known as “mirim”. This species belongs to the Humiriaceae family, and its tea is used in many Brazilian regions for its anti-inflammatory action, especially for treating uterine inflammation [22,23].

Analyses of the leaf extracts by HPLC-ESI-MS and FIA-ESI-IT/MS led to the identification of 11 substances, 10 of which had not yet been reported for this species: seven flavonoids (gallocatechin, kaempferol 3-neohesperidoside, sophoricoside, quercetin 3-arabinoside, quercetin-*O*-rhamnoside, kaempferol-dirhamnoside, and vitexin-dirhamnoside); one oligosaccharide (maltotetraose), bergenin; and two derivatives (galloylbergenin and hydroxybenzoyl bergenin).

The substances present in the chemical composition of *H. balsamifera* tell a lot about the species. Flavonoids, according to the characterization presented in this study, are the most abundant class of compounds. These substances exhibit high bioactive potential and present anti-ulcer, antioxidant, anti-inflammatory, anti-allergic, antitumor, antiviral, antifungal, and antidiabetic activities [34,35,36,37].

Antimicrobial tests with flavonoids have received increasing attention in recent years, since these compounds are synthesized by plants in response to various types of stress, including microbial infections [38,39,40]. Researchers are also interested in how flavonoids are able to exhibit antibacterial activity through mechanisms different from conventional drugs, hindering the development of microbial resistance [41,42].

During this study, the *H. balsamifera* extracts were subjected to three tests for the evaluation of their antimicrobial potential against different bacteria strains. By the end of the in vitro tests, the extracts which presented the best results were selected for the in vivo anti-infective assay using *T. molitor* larvae. Our results showed that the *H. balsamifera* ethyl acetate leaf extract (EALE) showed efficacy against *S. aureus*, one of the most resistant pathogens in existence, in all three tests (MIC, antibiofilm potential, and the tests in vivo). The efficacy of this extract is believed to be due to the flavonoids present in its composition. Flavonoids are well-known in the literature, as other polyphenols, to be able to inhibit microbial growth through several mechanisms, such as the inhibition of ATP synthesis in the electron transport chain, inhibition of nucleic acid synthesis, inhibition of the efflux pump, inhibition of biofilm formation, inhibition of virulence factors, inhibition of quorum sensing, membrane disruption, inhibitors of bacterial toxins, and inhibition of cell envelope synthesis [41,43,44].

Bergenin, identified in the three extracts analyzed, has already been reported in *H. balsamifera* [24] and other two species from the Humiriaceae family: *Endopleura uchi* and *Sacoglottis gabonensis* [45,46]. This isocoumarin and its derivatives, such as the identified flavonoids, can be directly related to the antimicrobial activity of the extracts against *S. aureus*. A recent study showed that six synthetic derivatives of bergenin obtained by Williamson synthesis inhibited *S. aureus* growth, especially 8,10-dihexyl-bergenin and 8,10-didecyl-bergenin, which presented the most promising MIC value: 3.12 µg/mL [47].

Besides antimicrobial activity, bergenin has also been pointed out as one of the main substances responsible for antimalarial [24], anti-inflammatory [48,49], antinociceptive [50], anxiolytic [51], and antioxidant activities [52,53].

## 4. Materials and Methods

### 4.1. Botanical Material

The leaves and stem bark of *Humiria balsamifera* (Aubl) were collected during the rainy season in Contrato Village, located in the municipality of Morros—MA (2°55′14.8″ S 43°55′38.8″ W). The material was identified at the Rosa Mochel Herbarium (SLUI) of the State University of Maranhão—UEMA, where an exsiccate was deposited under registration number 4769.

### 4.2. Preparation of the Extracts

The leaves were dried at room temperature, while the stem barks were dried at 40–50 °C in an oven for 24 h; after which, they were ground in a knife mill separately. The crude extracts were prepared from the ground materials by cold percolation, and each extraction was performed twice for each solvent within a period of 5 days, following the sequence hexane, ethyl acetate, and methanol. The extracts obtained were filtered and concentrated on a rotary evaporator [54].

### 4.3. Test Microorganisms

The microbial strains used in this work were obtained from the Microbial Culture Collection of CEUMA University. It used the following strains: *Escherichia coli ATCC 25922, Listeria monocytogenes* ATCC 15313, *Salmonella enterica* Typhimurium ATCC 14028, and *Staphylococcus aureus ATCC 6538.*

### 4.4. Minimum Inhibitory Concentration (MIC)

The antimicrobial activity of the extracts was evaluated by the determination of the minimum inhibitory concentration (MIC). The MIC was determined by the broth microdilution method. Sterile 96-well plates were prepared with 150 µL of Müeller Hinton broth (MHB) and 50 µL of the extract following the serial dilution method. After the dilutions (1250 µg/mL−2.0 µg/mL for the stem bark extracts and 6250 µg/mL–1.0 µg/mL for the leaf extracts), 10 µL of the microbial suspension were added to the plates until a 0.5 McFarland standard was reached, and they were then incubated in a lab oven at 37 °C for 24 h. The test was performed in duplicate. Once the incubation period ended, 20 µL of resazurin (Sigma-Aldrich, St. Louis, MO, USA; 0.03%) were added, and the readings were executed after 40 min of incubation at 37 °C. Alterations in color, from blue to pink, were considered an indication of microbial growth. The MIC was defined as the lowest concentration able to prevent microbial growth [55].

### 4.5. Antibiofilm Test

To evaluate the antibiofilm activity, a sample of 10 μL of *S. aureus* suspension (prepared as described in the MIC section) was mixed with 140 μL of MHB and 50 μL of EACH to reach subinhibitory concentrations (0.5 × MIC, 0.25 × MIC, 0.125 × MIC, and 0.0625 × MIC). After 24 h of incubation at 37 °C, the formed biofilm was fixed with methanol (P.A.), stained with violet crystal (0.1%), and washed with ethanol (P.A.). The biofilm mass was measured using a spectrophotometer at 550 nm. The absorbances of the wells that each received only the culture medium and bacterial solution were used as the positive control.

### 4.6. Infection Model Using Tenebrio molitor Larvae

The assessment of the antimicrobial effect in vivo used larvae of the insect *T. molitor* (Tenebrionidae). Larvae of approximately 100 mg were randomized into groups with a minimum of 10 individuals. Before inoculation, the cuticles were cleaned with 70% alcohol. An aliquot of 10 µL of the microbial suspension (1.0 × 10^11^ CFU/mL) was injected in the membrane region between the penultimate and antepenultimate rings of the larvae, which were then incubated at 37 °C. After 2 h, the larvae groups received 10 µL of each extract at different concentrations. Viability was assessed daily by the absence of movement. Larvae inoculated with the microorganism and treated with PBS were used as the negative control, while the noninfected larvae were selected as the positive control. Death of all larvae or transition into pulp form in the experimental group determined the end of the experiment [55].

### 4.7. Extracts Characterization by HPLC-ESI-MS and FIA-ESI-IT/MS

For the HPLC-ESI-IT/MS/MS and FIA-ESI-IT/MS^n^ analyses, a clean-up step was performed to remove any contaminants; the solution was purified by solid-phase extraction (SPE) using Phenomenex Strata C18 cartridges (500 mg of stationary phase) that were previously activated with 5 mL of MeOH and equilibrated with 5 mL of MeOH:H_2_O (1:1, *v*/*v*). The compounds were eluted from the cartridges using 1 mL of MeOH:H_2_O (1:1, *v*/*v*) with a final volume of 5 mL. The samples were then filtered using a 0.22-µm PTFE filter and dried. The extract was diluted to 10 µg/mL in the HPLC solvent, and then, aliquots of 20 µL were injected directly into the LC-ESI-IT/MS [56].

The analysis was carried out on an online LC-ESI-IT-MS in a LCQ Fleet Ion Trap Mass Spectrometer, Thermo Scientific^®^ (Waltham, MA, EUA). A Kinetex^®^ (Torrance, CA, USA) C18 LC column (2.1 × 100 mm, 100 Å, and 5 μm) was used to separate the components. The analysis was executed using water with formic acid 0.1% (A) and acetonitrile + formic acid 0.1% (B), with formic acid 0.1% added in gradient boosting, going from 10% to 100% in 6 min with a flow rate of 0.4 mL/min. The sample was injected into the HPLC system, where it was analyzed online by ESI-MS in the negative ion mode with a UV detector. Mass spectrometry was performed in an LCQ Fleet Ion Trap LC/MSⁿ—Thermo Scientific^®^ (Waltham, MA, EUA) [56].

A FIA-ESI-IT-MS^n^ flow injection analysis was performed in an LTQ XL™ linear ion trap mass spectrometer with an ESI ion source (electrospray ionization) in negative mode (Thermo, San Jose, CA, USA), using a stainless-steel capillary tube at 280 °C, 5.00 kV, capillary voltage of −90 V, −100 V tubular lens, and a flow rate of 5 μL min^−1^. A full-scan analysis was conducted at 100–1000 *m*/*z*. Multiple-stage fragmentations by electrospray ionization/multi-stage mass spectrometry (ESI-MSn) were performed using collision-induced dissociation (CID) by helium for ionic activation. The first step was a full-scan MS to acquire data from the ions at the selected *m*/*z*. The second step was a tandem mass spectrometry (MS/MS) in data-dependent acquisition mode on [M−H]^−^ molecules of the compounds of interest at a collision energy of 30% and activation time of 30 ms. The product ions were subjected to further fragmentation under the same conditions, until no more fragments were observed. The identification of different compounds in the chromatographic profile of the hydroalcoholic extract was carried out by comparing their retention times and UV spectra with the literature data [56].

### 4.8. Statistical Analysis

All tests were performed in at least two independent assays in quadruplicate. The statistical analyses were performed using the software GraphPad Prism version 5.01 (GraphPad Software Inc., La Jolla, CA, USA). Data were analyzed by two-way analysis of variance (ANOVA), followed by Tukey’s test. A *p*-value of < 0.05 was considered statistically significant.

## 5. Conclusions

The number of bioactive compounds found in the chemical composition of *H. balsamifera* emphasized its significance in both traditional medicine and scientific research of studies with new treatments based on substances from the Brazilian flora. This highlighted the importance of this study, since the analyses by HPLC-ESI-MS and FIA-ESI-IT/MS identified 11 substances, 10 of which had not yet been reported for *H. balsamifera*, improving the literature regarding its composition. Additionally, the antimicrobial activity of the ethyl acetate leaf extract (EALE) against *S. aureus* was, for the first time, described using three different strategies (MIC, antibiofilm activity, and tests in vivo), highlighting all the potential of this plant against one of the most resistant bacteria of the present day, which encourages further studies on natural bioactive metabolites that can be isolated from *H. balsamifera*.

## Figures and Tables

**Figure 1 plants-10-01479-f001:**
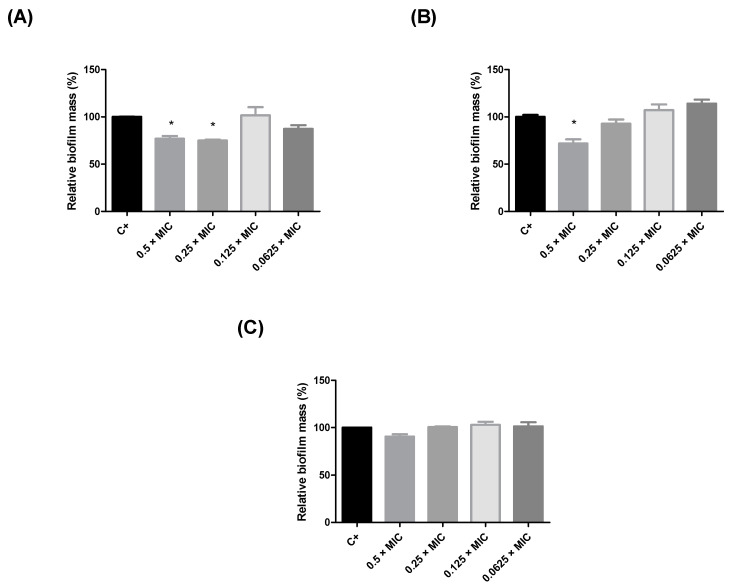
Inhibition of the biofilm formation in *Staphylococcus aureus* by *Humiria balsamifera* (Aubl) extracts. (**A**) HLE = hexane leaf extract, (**B**) EALE = ethyl acetate leaf extract, and (**C**) MLE = methanol leaf extract. (*) Significant differences (*p* < 0.05) in relation to untreated biofilm (C +).

**Figure 2 plants-10-01479-f002:**
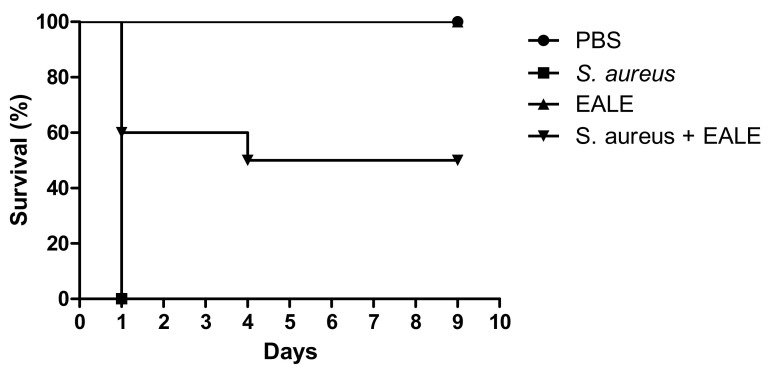
Effect of the *Humiria balsamifera* (Aubl) EALE treatment on the lifespan of *Tenebrio molitor* larvae during infection by *S. aureus* ATCC 6538. The larvae were treated with EALE at a dose of 3.12 mg/kg. Negative control groups injected with PBS (phosphate-buffered saline) were also included.

**Figure 3 plants-10-01479-f003:**
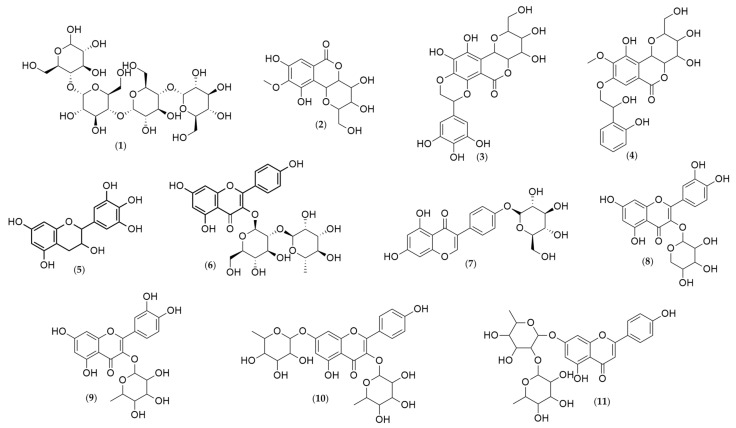
Chemical structures of the substances identified in the extracts of *Humiria balsamifera* (Aubl) by HPLC-ESI-IT/MS using the MassBank Spectral DataBase. Maltotetraose (**1**); Bergenin (**2**); Galloylbergenin (**3**); Hydroxybenzoyl bergenin (**4**); Gallocatechin (**5**); Kaempferol 3-*O*-neohesperidoside (**6**); Sophoricoside (**7**); Quercetin 3-arabinoside (**8**); Quercetin-*O*-rhamnoside (**9**); Kaempferol-dirhamnoside (**10**); and Vitexin-dirhamnoside (**11**).

**Table 1 plants-10-01479-t001:** Determination of the minimum inhibitory concentrations (MIC) of the leaf and stem bark extracts of *Humiria balsamifera* (Aubl).

Bacteria Species	HSBE	EASBE	MSBE	HLE	EALE	MLE
*E. coli ATCC 25922*	>12,500	>12,500	>12,500	>12,500	>12,500	>12,500
*L. monocytogenes ATCC 6538*	>12,500	>12,500	>12,500	>12,500	3120	3120
*S. aureus ATCC 6538*	>12,500	>12,500	>12,500	1560	780	780
*S. enterica Typhimurium ATCC 14028*	>12,500	>12,500	>12,500	>12,500	>12,500	>12,500

Footnote: HSBE = hexane stem bark extract, EASBE = ethyl acetate stem bark extract, MSBE = methanol stem bark extract, HLE = hexane leaf extract, EALE = ethyl acetate leaf extract, and MLE = methanol leaf extract. MIC values are expressed in µg/mL.

**Table 2 plants-10-01479-t002:** Identification of the substances present in the ethyl acetate stem bar extract of *Humiria balsamifera* (Aubl).

RT (min)	[M-H]	MSn	Proposed Substance	Reference
1.97	665	664, 305	Maltotetraose	[28]
2.85	327	249, 234, 207	Bergenin	[29]
3.58	479	327, 249, 234, 207	Galloylbergenin	[29]
4.87	463	327, 249, 234, 207	Hydroxybenzoyl bergenin	[29]

**Table 3 plants-10-01479-t003:** Identification of the substances present in the *Humiria balsamifera* (Aubl) methanol stem bark extract.

RT (min)	[M-H]	MSn	Proposed Substance	Reference
3.01	305	179, 165	Gallocatechin	[30]
3.81	327	234, 207, 192	Bergenin	[29]
6.35	593	431, 411	Kaempferol 3-neohesperidoside	[31]

**Table 4 plants-10-01479-t004:** Identification of the substances present in the *Humiria balsamifera* (Aubl) ethyl acetate leaf extract.

RT (min)	[M-H]	MSn	Proposed Substance	Reference
2.83	327	249, 234, 207, 192	Bergenin	[29]
5.27	431	311, 283, 341	Sophoricoside	[32]
6.09	433	300, 271, 243, 227	Quercetin 3-arabinoside	[33]
6.46	447	300, 271, 243	Quercetin-*O*-rhamnoside	[33]
7.17	577	431, 413, 341, 300	Kaempferol-dirhamnoside	[33]
8.38	561	431, 293	Vitexin-dirhamnoside	[32]

## Data Availability

Not applicable.

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
