# Peer review of "Antimicrobial Activity of Extracts from the Humiria balsamifera (Aubl)"

_plants, 2021, doi:10.3390/plants10071479_

Round 1
Reviewer 1 Report
The document describes the preparation, chemical analysis, and evaluation of the biological activity of Humiria balsamifera.
The subject is interesting and worth investigating. Moreover, the selection of bacterial strains for the evaluation of antimicrobial activity is also well defended.
Still, there are suggestions to improve the manuscript.
In the introduction, there is a focus on the need for new antimicrobials. Still, the study is related to evaluating the plant's polyphenols, and the selection of these chemical constituents is not justified.
How do you know that there are no terpenes or other small-sized molecules present in the extracts that can account for the biological activity? Please explain and include in the manuscript in the discussion.
The chemical analysis section is well prepared and complete, but the evaluation of the biological activity needs to be revised. Many arguments are not sufficiently supported. As an example, the authors claim that S. aureus is one of the most dangerous bacterial types. Still, you have P. aeruginosa and other strains that are even more pathogenic, but evaluating the biological activity against this pathogen was not effective.
The search for new antimicrobials from plant extracts is of particular importance given the worldwide antibiotic resistance crisis. However, the active concentration of the extract to further study their applications as new antibiotics is lower than what is presented by the authors. Therefore, I would suggest changing the focus of the introduction, not necessary for new antibiotics, but in search of antimicrobials that can be used in different applications, such in the food industry or in agroindustrial applications.
Methodology
The ATCC strain numbers for L. monocytogenes and S. enterica are missing.
The method for biofilm formation inhibition is different from what usually is done, where the EPS from the biofilm is first fixed with methanol and then stained. The reference given in the article also uses this method. Please explain or eliminate this section.
Author Response
Revision of the manuscript plants-1274771
Dear Editor,
Please find enclosed the revised version of our manuscript, plants-1274771, entitled “Antimicrobial activity of extracts from the Humiria balsamifera (Aubl)”. We are pleased to submit the revised version (alterations highlighted in red), and provide our responses to the reviewers below. The comments and suggestions have made a significant contribution to improving the paper. In addition, the text has been completely revised in the light of the very constructive comments made by the referees. We would like to take this opportunity to thank them for the excellent revision work.
#Reviwer 1
The subject is interesting and worth investigating. Moreover, the selection of bacterial strains for the evaluation of antimicrobial activity is also well defended. Still, there are suggestions to improve the manuscript.
Answer: We thank the reviewers for their comments.
In the introduction, there is a focus on the need for new antimicrobials. Still, the study is related to evaluating the plant's polyphenols, and the selection of these chemical constituents is not justified.
Answer: Thanks for the suggestion. The text has been changed in manuscript
How do you know that there are no terpenes or other small-sized molecules present in the extracts that can account for the biological activity? Please explain and include in the manuscript in the discussion.
Answer: We appreciate the reviewer's comment. The extracts were analyzed by liquid chromatography and mass spectrometry using electrospray ionization (ESI) source. The ESI source may not have ionized the compounds like steroids and triterpenes. We cannot claim that there are no such compounds in the extracts. The reviewer is right, we corrected the statement in the manuscript.
The search for new antimicrobials from plant extracts is of particular importance given the worldwide antibiotic resistance crisis. However, the active concentration of the extract to further study their applications as new antibiotics is lower than what is presented by the authors. Therefore, I would suggest changing the focus of the introduction, not necessary for new antibiotics, but in search of antimicrobials that can be used in different applications, such in the food industry or in agroindustrial applications.
Answer: Dear reviewer, thank you for this consideration.
Methodology
The ATCC strain numbers for L. monocytogenes and S. enterica are missing. The method for biofilm formation inhibition is different from what usually is done, where the EPS from the biofilm is first fixed with methanol and then stained. The reference given in the article also uses this method. Please explain or eliminate this section.
Answer: Dear reviewer, thank you for these considerations. We have added the ATCC numbers for the used strains in the appropriated sections. In relation to antibiofilm assays, we have corrected the method. In fact, we have used the standard assay based on biomass quantification by crystal violet staining. Sorry for this mistake.

Reviewer 2 Report
The article evaluates the antimicrobial activity of Humiria balsamifera on different species. the experimental part of the articles is strong, the metodology is adequate for the purpose of the article, the results are clearly presented. Congratulations! The discussion chapter should be improved, what can be the clincial applications of this activity on the biofilm? did any authors dicovered similar effects? a summarize table of previous studies would be importnat? are any formulas such as nanoparticles to deliver this substance? does this plant or its compound have any pleiotropic effects or interaction with oder drugs? Please consult the fallowing bibliography:
Sabir, F.; Barani, M.; Mukhtar, M.; Rahdar, A.; Cucchiarini, M.; Zafar, M.N.; Behl, T.; Bungau, S. Nanodiagnosis and Nanotreatment of Cardiovascular Diseases: An Overview. Chemosensors 2021, 9, 67. https://doi.org/10.3390/chemosensors9040067
Sabău, M. Real life anticoagulant treatment for stroke prevention in patients with nonvalvular atrial fibrillation. Farmacia 2020, 68,
912–918
Author Response
Revision of the manuscript plants-1274771
Dear Editor,
Please find enclosed the revised version of our manuscript, plants-1274771, entitled “Antimicrobial polyphenols in Humiria balsamifera (Aubl)”. We are pleased to submit the revised version (alterations highlighted in red), and provide our responses to the reviewers below. The comments and suggestions have made a significant contribution to improving the paper. In addition, the text has been completely revised in the light of the very constructive comments made by the referees. We would like to take this opportunity to thank them for the excellent revision work.
#Reviwer 2
The article evaluates the antimicrobial activity of Humiria balsamifera on different species. the experimental part of the articles is strong, the methodology is adequate for the purpose of the article, the results are clearly presented. Congratulations!
Answer: We thank the reviewers for their comments.
The discussion chapter should be improved, what can be the clinical applications of this activity on the biofilm? did any authors discovered similar effects? a summarize table of previous studies would be important? are any formulas such as nanoparticles to deliver this substance? does this plant or its compound have any pleiotropic effects or interaction with oder drugs?
Answer: Dear reviewer, thank you for these considerations. We use the antibiofilm assay as one of the strategies to assess antimicrobial activity, it is a very important assay in the search for new antimicrobials, in the future we intend to carry out more tests with isolated substances. So far, we did only antimicrobial activity assays with H. balsamifera extracts.

Reviewer 3 Report
The subject taken up by the authors is very interesting. Research on plant raw materials in terms of their antimicrobial activity has been very much developed recently. In studies, however, I lacked information on quantitative analysis. Have the authors tested the concentration of these substances? If so, please supplement the publication with such data. if not, please explain why. In the case of the presented extraction methods, other substances with a similar effect could be extracted together with the tested compounds. Please provide the full characteristics of the extracts. The reviewer noted that the MIC is at a very high level. Please explain in the discussion why such a high concentration of extracts is needed. Additionally, I have the following comments: Fig.2. please change the unit from mg/Kg to mg/kg Fig 3. Lack of description of the substances shown in the figure. please enter their names in the legend of Figures
Author Response
Revision of the manuscript plants-1274771
Dear Editor,
Please find enclosed the revised version of our manuscript, plants-1274771, entitled “Antimicrobial polyphenols in Humiria balsamifera (Aubl)”. We are pleased to submit the revised version (alterations highlighted in red), and provide our responses to the reviewers below. The comments and suggestions have made a significant contribution to improving the paper. In addition, the text has been completely revised in the light of the very constructive comments made by the referees. We would like to take this opportunity to thank them for the excellent revision work.
#Reviwer 3
The subject taken up by the authors is very interesting. Research on plant raw materials in terms of their antimicrobial activity has been very much developed recently. In studies, however, I lacked information on quantitative analysis. Have the authors tested the concentration of these substances? If so, please supplement the publication with such data. if not, please explain why.
Answer: Thanks for the question. We did not test the isolated compounds as our focus was to evaluate the extracts. But in a future study we intend to isolate those identified by LC-MS
In the case of the presented extraction methods, other substances with a similar effect could be extracted together with the tested compounds. Please provide the full characteristics of the extracts.
Answer. Yes, other substances with a similar effect can be have extracted, but the extracts were analyzed by liquid chromatography and mass spectrometry using electrospray ionization (ESI) source. The ESI source may not have ionized the compounds like steroids and triterpenes. We cannot claim that there are no such compounds in the extracts. But the chromatography analysis showed that phenolic compounds appear to be the majority in the extract.
The reviewer noted that the MIC is at a very high level. Please explain in the discussion why such a high concentration of extracts is needed.
Answer: Thanks for the suggestion. The text has been changed in manuscript
Além disso, tenho os seguintes comentários: Fig.2. mude a unidade de mg / Kg para mg / kg Fig 3. Falta de descrição das substâncias mostradas na figura. por favor insira seus nomes na legenda de figuras.
Answer: Thanks for the suggestion. The text has been changed in manuscript

Round 2
Reviewer 1 Report
The document has been substantially improved, and I thank the authors for considering my suggestions. Still, there are few corrections that need to be addressed.
The strain of Salmonella used needs to be homogenized to Salmonella enterica Typhimorium or only Salmonella Typhimorium. Since typhimorium is referred as to the serotype of the S. enterica species, does not need to be italicized. In the document, there are sections where it is referred as S. typhimorium or S. enterica.
In the abstract, it is always preferred that numerical values of the results are included. They were included in the original version but deleted in the revised version. Please correct.
Author Response
Revision of the manuscript plants-1274771
#Reviwer 1
The document has been substantially improved, and I thank the authors for considering my suggestions. Still, there are few corrections that need to be addressed.
Answer: We appreciate the reviewer's comment.
The strain of Salmonella used needs to be homogenized to Salmonella enterica Typhimorium or only Salmonella Typhimorium. Since typhimorium is referred as to the serotype of the S. enterica species, does not need to be italicized. In the document, there are sections where it is referred as S. typhimorium or S. enterica.
Answer: Dear reviewer, thank you for this suggestion. The text has been changed in manuscript.
In the abstract, it is always preferred that numerical values of the results are included. They were included in the original version but deleted in the revised version. Please correct.
Answer: Dear reviewer, thank you for this suggestion. The text has been changed in manuscript.
